# Methane seep carbonates yield clumped isotope signatures out of equilibrium with formation temperatures

S.J. Loyd[1,2], J. Sample[3], R.E. Tripati[2,4,5], W.F. Defliese[2], K. Brooks[2], M. Hovland[6], M. Torres[7], J. Marlow[8], L.G. Hancock[9], R. Martin[10], T. Lyons[9] & A.E. Tripati[2,4,5]

Methane cold seep systems typically exhibit extensive buildups of authigenic carbonate minerals, resulting from local increases in alkalinity driven by methane oxidation. Here, we demonstrate that modern seep authigenic carbonates exhibit anomalously low clumped isotope values ($\Delta_{47}$), as much as $\sim 0.2$‰ lower than expected values. In modern seeps, this range of disequilibrium translates into apparent temperatures that are always warmer than ambient temperatures, by up to 50 °C. We examine various mechanisms that may induce disequilibrium behaviour in modern seep carbonates, and suggest that the observed values result from several factors including kinetic isotopic effects during methane oxidation, mixing of inorganic carbon pools, pH effects and rapid precipitation. Ancient seep carbonates studied here also exhibit potential disequilibrium signals. Ultimately, these findings indicate the predominance of disequilibrium clumped isotope behaviour in modern cold seep carbonates that must be considered when characterizing environmental conditions in both modern and ancient cold seep settings.

[1] Department of Geological Sciences, California State University, Fullerton, California 92831, USA. [2] Department of Earth, Planetary and Space Sciences, University of California, Los Angeles, California 90095, USA. [3] School of Earth Sciences and Environmental Sustainability, Northern Arizona University, Flagstaff, Arizona 86001, USA. [4] Department of Atmospheric and Oceanic Sciences, Institute of the Environment and Sustainability, University of California, Los Angeles, California 90095, USA. [5] European Institute of Marine Sciences (IUEM), Université de Brest, UMR 6538/6539, Rue Dumont D'Urville, and IFREMER, Plouzané 29019, France. [6] Centre for Geobiology, University of Bergen, Bergen 5003, Norway. [7] College of Earth, Ocean, and Atmospheric Sciences, Oregon State University, Corvallis, Oregon 97331, USA. [8] Division of Geological and Planetary Sciences, California Institute of Technology, Pasadena, California 91125, USA. [9] Department of Earth Sciences, University of California, Riverside, California 92521, USA. [10] Department of Earth and Space Sciences/ Burke Museum, University of Washington, Seattle, Washington 98195, USA. Correspondence and requests for materials should be addressed to S.J.L. (email: sloyd@fullerton.edu) or to A.E.T. (email: atripati@g.ucla.edu).

Methane cold seeps host diverse macro and microbiological communities[1–5]. These ecosystems are driven by microbially mediated reactions involving methane-containing fluids advecting from depth[6]. As methane ascends it is primarily oxidized anaerobically by sulfate (termed the anaerobic oxidation of methane (AOM)), or aerobically in the presence of oxic seawater, as follows:

$$CH_4 + SO_4^{2-} \rightarrow HCO_3^- + HS^- + H_2O \qquad (1)$$

$$CH_4 + 2O_2 \rightarrow HCO_3^- + H_2O + H^+ \qquad (2)$$

Both reactions produce bicarbonate, but only methane oxidation coupled with sulfate reduction increases alkalinity. The alkalinity production fostered by eq. 1 sustained by the relatively high concentration of sulfate in seawater promotes extensive carbonate (and sulfide) mineral production near methane seeps[7]. Complex microbial consortia facilitate sulfate reduction-coupled methane oxidation[8] yielding extensive and generally rapid authigenic carbonate production in cold seep environments.

In an attempt to characterize the interactions in cold seep systems, both modern[4,9–12] and ancient[10,11,13–15] cold seep deposits have been studied extensively. Of particular importance is the characterization of ancient cold seep environmental conditions and their relationship to modern seep systems. However, due to the inherent issues associated with studies of past environments, characterization of ancient seeps hinges on the reliability of paleoproxies. A particularly useful parameter to quantify is precipitation temperature, as it dictates thermodynamic considerations such as abiotic versus biotic reaction times, and gas hydrate dynamics[16]. The newly emerging clumped isotope proxy has shown promise as a powerful geothermometer in the geosciences[17–22], yet the utility of clumped isotopes as an accurate geothermometer in cold seep carbonates has yet to be demonstrated.

Here, we demonstrate the occurrence of non-temperature dependent carbonate clumped isotope signatures in cold seep carbonates through analyses of modern precipitates forming under well-constrained conditions (that is, temperatures, pH, salinities and fluid $\delta^{18}O$ values). We compare modern systems with clumped isotope signatures in ancient seep carbonates, and discuss potential mechanisms to explain the observed disequilibrium values.

## Results

**Geologic context.** Both modern and ancient cold seep carbonates are explored here. Modern samples originate from Hydrate Ridge, offshore Costa Rica, the Eel River Basin and the Norwegian Sea. Ancient samples originate from the Tepee Buttes Colorado (Cretaceous), the Panoche Hills California (Paleocene), the Quinault Formation (Mio-Pliocene), the Pysht Formation (Eocene) and the Lincoln Creek Formation (Oligocene; Fig. 1).

Modern cold seeps occur along continental margins and within large inland seas (for example, the Black and Mediterranean Seas)[23,24]. These sites exhibit spatially and temporally variable delivery of subsurface methane, derived from sedimentary production (microbial or thermogenic), to relatively oxic marine waters[25]. This methane is oxidized microbially either by reaction with dissolved oxygen or sulfate[26], as explained above. Photographs, photomicrographs and previously reported and new $\delta^{13}C$ and $\delta^{18}O$ values are provided in Figs 2–4, respectively.

Cold seep systems of Hydrate Ridge exhibit authigenic carbonate buildups in the form of chimneys, crusts, slabs, cements and concretions[12,27–30]. These carbonate buildups are actively forming, and carbon-14 ages indicate that authigenesis has occurred within the last ∼40 ka (ref. 30). Carbonate $\delta^{13}C$ ($\delta^{13}C_{carb}$) values are extremely negative and commonly extend to

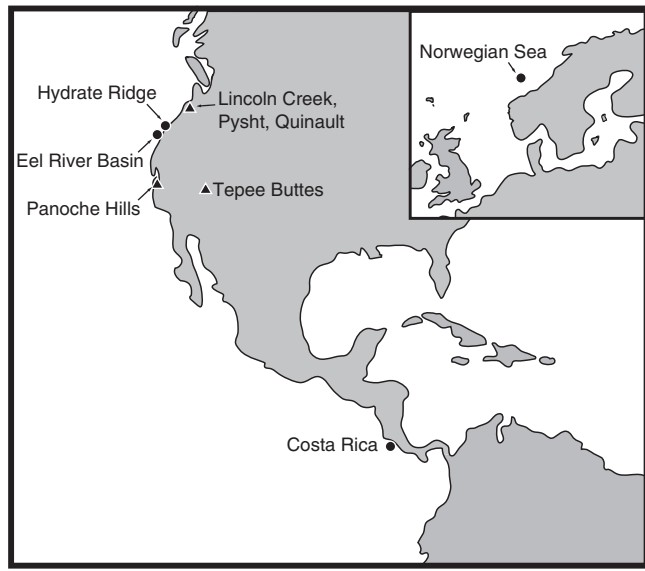

**Figure 1 | Map of sample sites.** Circles and triangles correspond to modern and ancient seep sites, respectively.

lesser than –30‰ (VPDB), indicative of a significant methane carbon source (Fig. 4). Carbonates consist of aragonite, calcite, high-magnesium calcite and dolomite primarily as micritic and acicular cements[31–33] (Figs 2 and 3). Bottom water temperatures of Hydrate Ridge are ∼4–5 °C (ref. 12).

Authigenic carbonates (Figs 2 and 3) associated with cold seeps of the Eel River Basin occur primarily as irregular carbonate slabs, cements and concretions[4]. Erosion has exposed ancient carbonates, such that slabs commonly crop out on the seafloor despite formation in shallow sediments[4]. Authigenic carbonates are composed of magnesium calcite, aragonite and dolomite[4]. Aragonite and calcite $\delta^{13}C$ and $\delta^{18}O$ values do not overlap with dolomites but instead exhibit $\delta^{13}C$ and $\delta^{18}O$ values (Fig. 4) that range from –40 to –3.2‰ and +3.2 and +5.8‰, respectively. In contrast, dolomites express significantly $^{13}C$-enriched values with $\delta^{13}C$ and $\delta^{18}O$ values ranging from +5.0 to +15‰ and +6.1 to +8.9‰, respectively[4]. The bottom water temperatures of Eel River Basin are ∼5 °C (ref. 4).

Cold seep sites off the coast of Costa Rica exhibit similar authigenic carbonate precipitates (Figs 2 and 3). These precipitates include concretions, carbonate 'clasts' and carbonate cemented muds[34]. Larger buildups are oftentimes referred to as chemoherm carbonates due to the direct association with characteristic cold seep fossil assemblages[9,32]. These chemoherm carbonates appear morphologically similar to slabs identified at the Hydrate Ridge and Eel River Basin localities. In addition, bottom water temperatures are ∼5 °C, similar to Hydrate Ridge and Eel River Basin[9].

The Norwegian Sea hosts cold seep systems yielding primarily aragonitic and calcitic carbonate buildups exposed in seafloor 'pockmarks'[35]. These buildups exhibit variable morphologies including crusts, ridges and blocky, tubular or irregular structures[33,36]. Carbonate carbon and oxygen ($\delta^{18}O_{carb}$) isotope values are ∼ −50 to −48‰ and ∼ +5 to +6‰, respectively[36,37] (Fig. 4), similar to values recognized at other modern cold seep sites. The bottom water temperatures are significantly colder than the other sites, ranging from ∼ −1 to 1 °C (ref. 35).

In addition to modern seep carbonates, samples of Cretaceous to Oligocene seep carbonates from the western USA are explored here. These carbonates are inferred to have cold seep affinity due to depleted $\delta^{13}C$ values (below −30‰) and/or the presence of

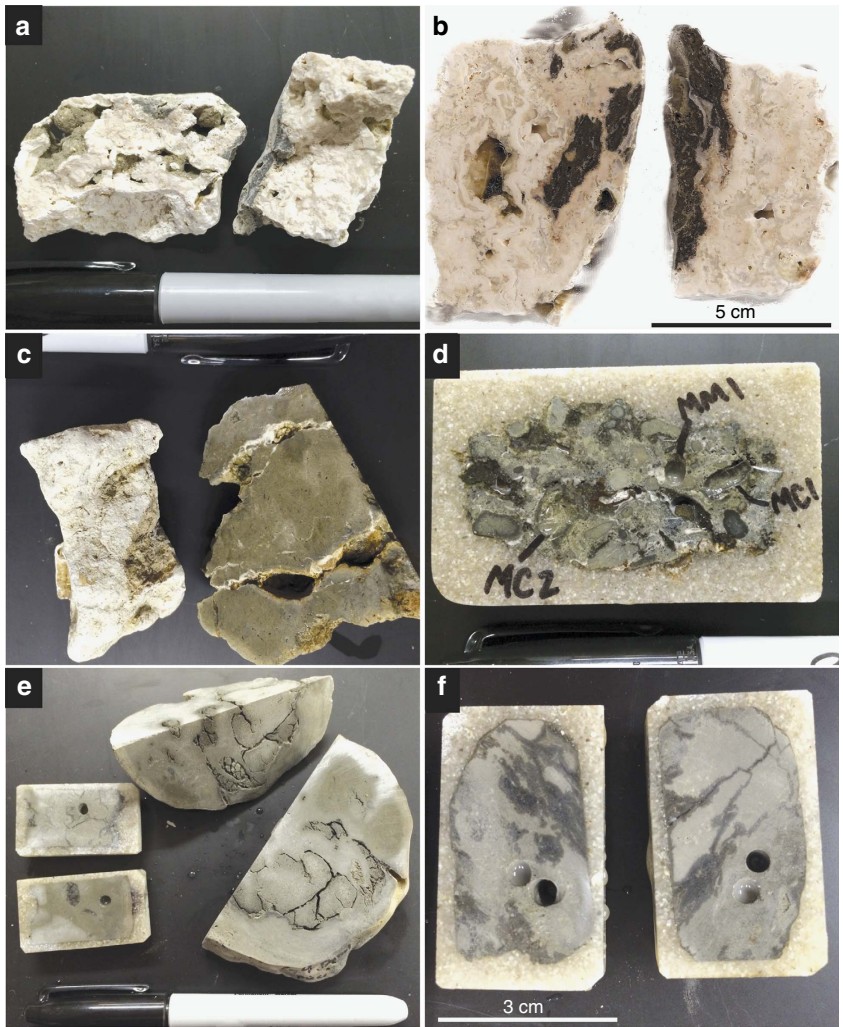

**Figure 2 | Photographs of cold seep carbonates.** (**a**,**b**) Hydrate Ridge samples exhibiting light and dark primarily carbonate phases. (**c**,**d**) Costa Rica cold seep carbonate with micrite (MM1) and clotted micrite (MC1,2) sample sites indicated. (**e**,**f**) Eel River Basin cold seep dolomite samples exhibiting similar light and dark colour variation as Hydrate Ridge samples. Permanent marker cap is 5-cm long.

diagnostic cold seep fossils. The ancient seep carbonates from these localities exhibit similar fabrics to modern seeps including fibrous and finely crystalline micrites[4]. In addition to these, coarser-crystalline phases including sparry vug filling cements and various sparites occur. Typically, these spars express [18]O-depleted and less [13]C-depleted compositions, respectively, potentially indicative of formation during later diagenesis (Fig. 4). Ultimately, ancient seeps record a more complex paragenetic evolution than modern seep carbonates[10,11,14,15,38].

**Modern cold seep carbonates**. Carbonate carbon isotope compositions are generally very low, whereas oxygen isotope values are elevated (Fig. 4). Specifically, $\delta^{13}C_{carb}$ values range from $-54.64$ to $-3.02‰$, $-49.86$ to $-39.00‰$, $+13.97$ to $+15.95‰$, $-52.01$ to $-50.82‰$ for carbonates from Hydrate Ridge, Costa Rica, the Eel River Basin and the Norwegian Sea, respectively. Respective $\delta^{18}O_{carb}$ values range from $+3.11$ to $+7.76‰$, $+3.76$ to $+4.10‰$, $+7.26$ to $+7.34‰$ and $+4.72$ to $+5.42‰$ for carbonates from Hydrate Ridge, Costa Rica, the Eel River Basin and the Norwegian Sea. These isotope compositions are consistent with values observed in these and similar cold seep carbonates reported elsewhere[4,11,12,27,37,39].

Clumped isotope compositions (Fig. 5, Tables 1 and 2) overlap among all of the modern sample sites. In addition, clumped

isotope compositions of fibrous and micritic carbonates (the two most abundant phases recognized) do not exhibit crystal habit-related differences in $\Delta_{47}$. The $\Delta_{47}$ values range from 0.620–0.679‰, 0.639–0.677‰, 0.706–0.740‰ and 0.705–0.783‰ for carbonates from Hydrate Ridge, Costa Rica, the Eel River Basin and the Norwegian Sea, respectively. Equilibrium values calculated for 1 and 5 °C bottom waters are 0.790 and 0.771‰, respectively, using a steep slope calibration[40], and 0.768 and 0.753‰, respectively, using a shallow slope calibration[41]. These calibrations were selected because both were generated partially or fully on the same instrument used for measuring the samples analysed for this study. The extent of disequilibrium (offset from expected equilibrium values) is $\sim +0.014$ to $-0.134‰$ in the most conservative scenario, and up to $-0.008$ to $-0.152‰$ (Table 3), yielding temperatures that are always warmer than ambient temperatures. These clumped isotope compositions translate to reconstructed temperatures ranging from 3–44 °C when a steep slope calibration is used[40]. If a shallow slope calibration[41] is used, the reconstructed temperatures range from $-3$ to 52 °C.

**Ancient cold seep carbonates**. Traditional $\delta^{13}C$ and $\delta^{18}O$ values show a wider range in ancient seep carbonates (Fig. 4). Carbon isotope values range from $-45.21$ to $-14.31‰$, $-30.08$ to

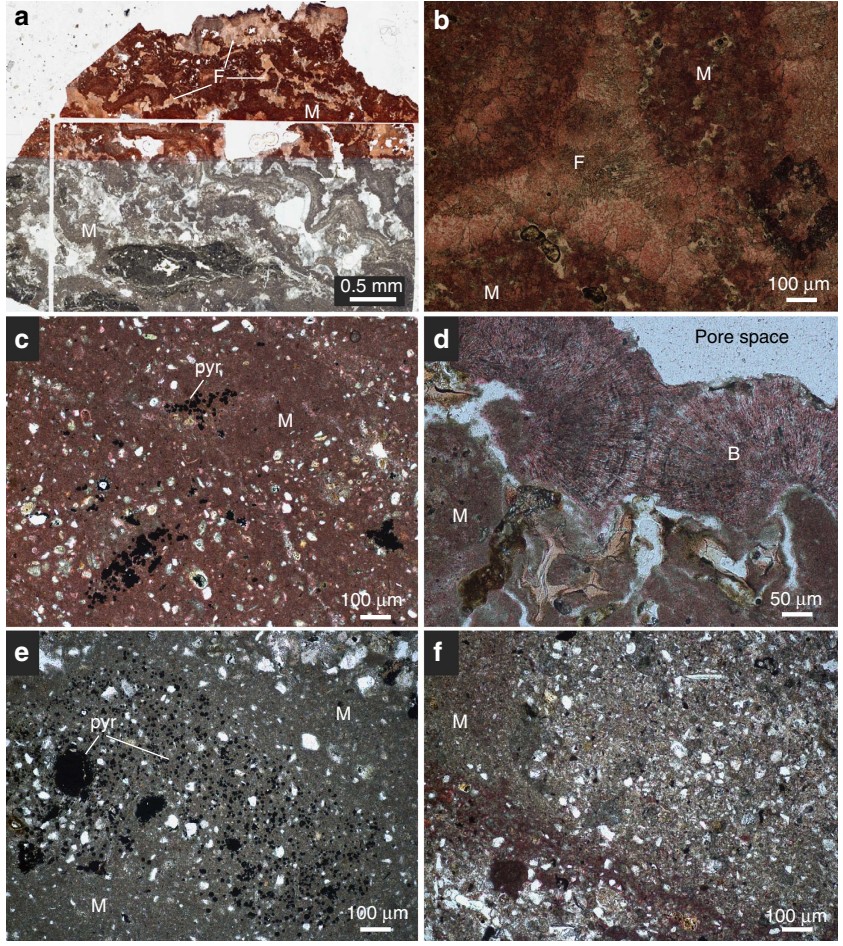

**Figure 3 | Photomicrographs of cold seep carbonates.** Red stain indicates calcium carbonate phases. Micritic (M), fibrous (F) and botryoidal (B) cements occur. (**a,b**) Hydrate Ridge samples composed of nearly pure calcium carbonate with a substantial fibrous component. (**c**) Costa Rica micritic carbonate with pyrite accumulations (pyr). (**d**) Micritic and botryoidal cold seep carbonate of Costa Rica, pale blue regions indicate pore space. White angular grains predominantly composed of quartz. (**e,f**) Eel River Basin largely micritic carbonates. The lack of stain in Eel River Basin sections indicates that these samples are predominantly composed of dolomite. Disseminated pyrite and pyrite accumulations occur in **e**. As in **c**, white grains in **e** and **f** composed of quartz.

$-6.05‰$, $-19.33$ to $-4.03‰$ and $-34.99$ to $-0.92‰$ in cold seep carbonates of the Tepee Buttes, Quinault Formation, Pysht Formation and Panoche Hills, respectively (Table 2). Oxygen isotope compositions range from $-12.09$ to $-0.30‰$, $-8.32$ to $+0.91‰$, $-1.00$ to $+1.53‰$ and $-0.71$ to $+2.25‰$ in cold seep carbonates of the Tepee Buttes, Quinault Formation, Pysht Formation and Panoche Hills, respectively (Table 2). The Lincoln Creek Formation carbonate exhibits $\delta^{13}C_{carb}$ and $\delta^{18}O_{carb}$ values of $-15.16$ and $+1.25‰$, respectively (Table 2). These $\delta^{13}C_{carb}$ and $\delta^{18}O_{carb}$ values broadly overlap with those reported previously[11,13,14,38,42,43] although some of the Tepee Buttes values are lower (Fig. 4).

Ancient cold seep carbonates exhibit clumped isotope compositions that range from those expressed by modern precipitates to significantly lower $\Delta_{47}$ values (Fig. 5; Table 2). Clumped isotope signatures from micritic, fibrous and sparry phases of the Tepee Buttes show no obvious crystallographic-specific distributions (that is, values generally overlap among phases). Tepee Buttes, Quinault Formation, Pysht Formation and Panoche Hills $\Delta_{47}$ values range from 0.377–0.521‰, 0.558–0.644‰, 0.611–0.631‰ and 0.649–0.705‰, respectively. A single sample form the Lincoln Creek Formation yielded a value of 0.623‰. Using the calibration by Tang et al.[41], these

clumped isotope values translate to temperatures ranging from 107–286 °C, 42–83 °C, 47–56 °C, 19–40 °C and 46 °C, respectively (Supplementary Table 1). The calibration by Tripati et al.[40], calibration yields temperatures ranging from 81–169 °C, 37–66 °C, 41–47 °C, 20–35 °C and 40 °C, respectively (Supplementary Table 1).

## Discussion

The $\delta^{13}C_{carb}$ values reported here from Hydrate Ridge, Costa Rica and the Norwegian Sea are consistent with carbonate formation from dissolved inorganic carbon generated via methane oxidation[4]. In contrast, the positive $\delta^{13}C_{carb}$ values of the Eel River Basin carbonates indicate formation in sediments experiencing microbial methane production[44]. The $\delta^{18}O_{carb}$ values of cold seep carbonates are $\sim +3$ to $+5‰$ higher than those expected for carbonates precipitated in normal seawater with a $\delta^{18}O$ of $\sim 0‰$ (Fig. 4). These enriched oxygen isotope values may reflect gas hydrate (clathrate) dissolution at depth, the presence of which has been established at most of the study sites[4,9,12,37].

The clumped isotope values presented in this study are not in isotopic equilibrium with ambient temperatures of precipitation.

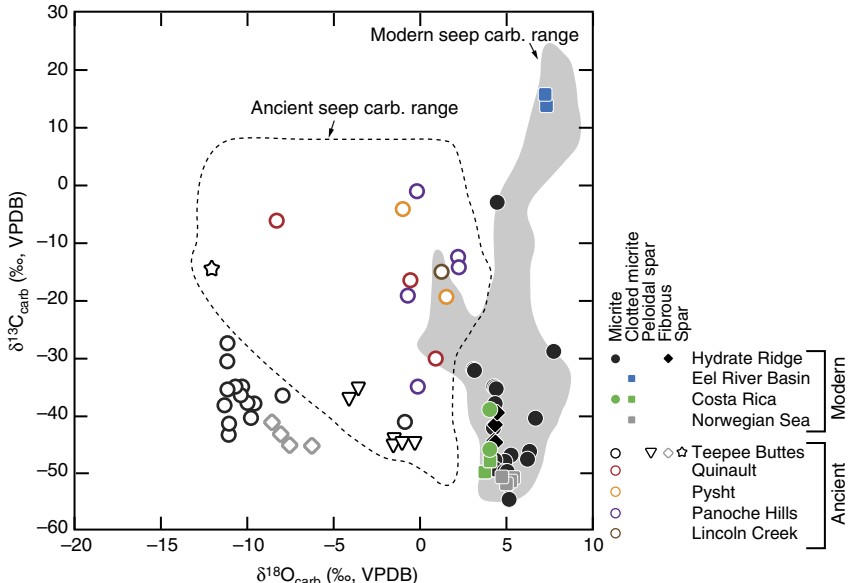

**Figure 4 | Carbon and oxygen isotope values of modern and ancient cold seep carbonates.** Ranges of previously reported values denoted by the grey envelope (modern[4,11,12,32,37,39]) and dashed outline (ancient[11,38,42]). Notice predominance of depleted $\delta^{13}C_{carb}$ values, typical of AOM carbonates. The enriched $\delta^{18}O_{carb}$ values expressed by modern cold seep carbonates has been interpreted to reflect incorporation of clathrate-dissolution-derived oxygen. Eel River Basin samples (micritic dolomites) exhibit positive $\delta^{13}C_{carb}$ values, indicative of a methanogenesis origin.

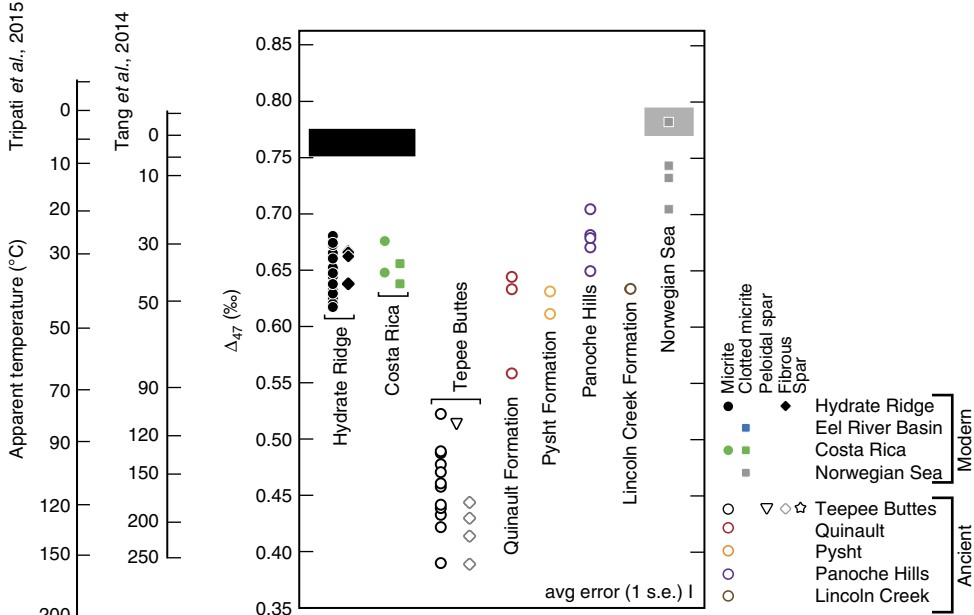

**Figure 5 | Clumped isotope compositions and reconstructed temperatures.** Also included are bottom water temperature ranges of the modern sites explored here. The grey envelope corresponds to Norwegian Sea and the black envelope corresponds to Hydrate Ridge and Costa Rica.

The sediment-water interface temperature at each site is $\sim$0–5 °C (refs 4,12,34,37). The $\Delta_{47}$ values are on the order of $\sim$0.1‰ lower than expected for an equilibrium precipitate (Fig. 5, Table 3). This translates to significantly elevated carbonate formation temperatures (up to 52 °C) compared with the ambient conditions and well beyond reasonable annual temperature variations of bottom waters. In some cases, authigenic carbonates associated with cold seeps have been postulated to grow after burial and then be exhumed by later sediment winnowing. Whereas this process may account for re-exposure after shallow ($\ll$50 m) burial, it seems very unlikely that winnowing would remove the 100s of metres of the sediment required to reach depths heated to $\sim$25 °C via burial. Of course, deep burial would be unnecessary if warm fluids were advecting from depth. Modern seep fluids are not hot enough to perturb the ambient bottom water temperatures, however, one could argue that fluid temperatures differed in the recent past. Ultimately, carbonate precipitation at elevated temperature is unlikely given the $\delta^{18}O_{carb}$ data, which are enriched compared with modern seawater and furthermore exhibit bulk isotope absolute values

**Table 1 | Modern seep carbonate geochemical data**

| Sample | Phases | Modern Temp (°C) | $\delta^{13}C_{carb}$ (‰,VPDB) | stdv (‰) | $\delta^{18}O_{carb}$ (‰,VPDB) | s.d. (‰) | $\Delta_{47}$ (‰, ARF) AFF 0.000 | $\Delta_{47}$ (‰, ARF) AFF 0.092 | s.e. (‰) | s.d. (‰) | n |
|---|---|---|---|---|---|---|---|---|---|---|---|
| Hydrate Ridge | | | | | | | | | | | |
| Depth: 690 m; position: 44.6722°N, 125.1219°W | | | | | | | | | | | |
| 2282-1c | Micrite | 5 | − 52.07 | 0.010 | 5.10 | 0.024 | | 0.632 | 0.013 | | 1 |
| 2282-3 | Micrite | 5 | − 28.83 | 0.008 | 7.76 | 0.0215 | | 0.666 | 0.0130 | | 1 |
| Depth: 630 m; position: 44.6694°N, 125.0480°W | | | | | | | | | | | |
| 2284-1 | Micrite | 5 | − 47.57 | 0.008 | 6.27 | 0.013 | | 0.661 | 0.012 | 0.028 | 2 |
| 2284-10 | Micrite | 5 | − 52.00 | 0.007 | 5.08 | 0.014 | | 0.679 | 0.011 | | 1 |
| 2284-2 | Micrite | 5 | − 46.98 | 0.005 | 5.28 | 0.009 | | 0.663 | 0.012 | 0.034 | 2 |
| 2284-3 | Micrite | 5 | − 46.25 | 0.005 | 6.36 | 0.0149 | | 0.643 | 0.0100 | | 1 |
| 2284-4 | Micrite | 5 | − 48.07 | 0.007 | 4.95 | 0.011 | | 0.649 | 0.014 | | 1 |
| 2284-5 | Micrite | 5 | − 40.48 | 0.006 | 6.70 | 0.0191 | | 0.672 | 0.0090 | | 1 |
| 2284-6 | Micrite | 5 | − 54.64 | 0.005 | 5.16 | 0.016 | | 0.659 | 0.013 | 0.039 | 2 |
| 2284-7 | Micrite | 5 | − 49.76 | 0.004 | 5.00 | 0.009 | | 0.647 | 0.021 | 0.032 | 2 |
| 2284-8 | Micrite | 5 | − 3.02 | 0.001 | 4.44 | 0.0081 | | 0.647 | 0.0294 | | 1 |
| 2284-9 | Micrite | 5 | − 31.97 | 0.006 | 3.11 | 0.0121 | | 0.636 | 0.0047 | | 1 |
| Depth: 800 m; position: 44.57°N, 125.15°W | | | | | | | | | | | |
| CS-1-B D1 | Micrite | 5 | − 49.53 | 0.003 | 4.35 | 0.007 | | 0.662 | 0.010 | | 1 |
| CS-1-B D2 | Micrite | 5 | − 49.38 | 0.002 | 4.28 | 0.005 | | 0.660 | 0.005 | | 1 |
| CS-1-B L2 | Micrite | 5 | − 35.26 | 0.003 | 4.34 | 0.007 | | 0.620 | 0.009 | | 1 |
| CS-1-B-L1 | Micrite | 5 | − 37.92 | 0.002 | 4.38 | 0.009 | | 0.663 | 0.006 | | 1 |
| cs-1-c d1 | Micrite | 5 | − 47.83 | 0.002 | 4.36 | 0.005 | | 0.629 | 0.007 | | 1 |
| CS-1-C L1 | Micrite | 5 | − 35.81 | 0.004 | 4.51 | 0.004 | | 0.650 | 0.013 | | 1 |
| CS-1-C L2 | Micrite | 5 | − 34.97 | 0.007 | 4.28 | 0.012 | | 0.644 | 0.010 | | 1 |
| HR-4635 3364-L-1 | Micrite | 5 | − 42.16 | 0.006 | 4.26 | 0.016 | | 0.652 | 0.021 | | 1 |
| CS-1-C A1 | Fibrous cement | 5 | − 44.63 | 0.003 | 4.41 | 0.007 | | 0.667 | 0.007 | | 1 |
| CS-1-C A2 | Fibrous cement | 5 | − 44.36 | 0.003 | 4.32 | 0.007 | | 0.638 | 0.013 | | 1 |
| CS-1-C-A3 | Fibrous cement | 5 | − 39.42 | 0.005 | 4.45 | 0.018 | | 0.663 | 0.011 | | 1 |
| HR-4635 3364-A-1 | Fibrous cement | 5 | − 41.52 | 0.002 | 4.37 | 0.005 | | 0.666 | 0.012 | | 1 |
| Eel River Basin | | | | | | | | | | | |
| Depth: 520 m; position: 40.812°N, 124.612°W | | | | | | | | | | | |
| ERB-4256-G4-A | Clotted micrite | 5 | 13.97 | 0.005 | 7.34 | 0.016 | | 0.740 | 0.010 | | 1 |
| ERB-4256-G4-B | Clotted micrite | 5 | 15.95 | 0.005 | 7.26 | 0.013 | | 0.706 | 0.008 | | 1 |
| Costa Rica | | | | | | | | | | | |
| Depth: 1,000 m; position: 8.93°N, 84.30°W | | | | | | | | | | | |
| CR-4501-L1-CM1 | Clotted micrite | 5 | − 49.86 | 0.003 | 3.76 | 0.013 | | 0.639 | 0.011 | | 1 |
| CR-4501-L1 CM 2 | Clotted micrite | 5 | − 47.84 | 0.004 | 4.07 | 0.004 | | 0.657 | 0.016 | | 1 |
| CR-4502-S3 MC 1 | Micrite clast | 5 | − 45.98 | 0.002 | 4.06 | 0.004 | | 0.677 | 0.009 | | 1 |
| CR 4502-L2-MC2 | Micrite clast | 5 | − 39.00 | 0.002 | 4.10 | 0.007 | | 0.649 | 0.009 | | 1 |
| Norwegian Sea | | | | | | | | | | | |
| Depth: 750 m; position: 64.67°N, 5.29°E | | | | | | | | | | | |
| H03 (90C) | Clotted micrite | 1 | − 50.82 | 0.005 | 4.72 | 0.012 | | 0.705 | 0.032 | | 1 |
| H03 (25C) | Clotted micrite | 1 | − 50.97 | 0.004 | 5.42 | 0.009 | 0.744 | | 0.021 | 0.051 | 2 |
| H04 (90C) | Clotted micrite | 1 | − 52.01 | 0.002 | 4.98 | 0.006 | | 0.733 | 0.023 | | 1 |
| H04 (25C) | Clotted micrite | 1 | − 51.57 | 0.881 | 5.24 | 0.395 | 0.783 | | 0.017 | 0.018 | 2 |

consistent with equilibrium precipitation from ambient fluids. Therefore, we reject the possibility that the majority of these carbonates formed at temperatures consistent with the predictions based on the clumped isotope data.

Several factors are likely to be important in contributing to the observed clumped isotope disequilibrium signatures at methane seeps. We hypothesize that the AOM reaction (eqs. 1 and 2) produces a kinetic isotope signal in the clumped isotope composition of the generated DIC species (Fig. 6). Kinetic effects have been shown to result from $CO_2$ hydration and hydroxylation reactions (Fig. 6)[17,45–47], and dehydration/dehydroxylation reactions[45]. Similar effects also occur in association with oxygen and carbon isotope partitioning among the gaseous and dissolved inorganic carbon phases[48]. It is highly likely that rate-limiting

steps may occur during the methane oxidation process, which may be complex as AOM involves multiple intermediate reaction steps[25]. We predict these rate-limiting steps may induce a large disequilibrium composition in product $HCO_3^-$ or $CO_3^{2-}$, as with these other systems. Modern cold seep carbonate $\Delta_{47}$ values suggest that AOM leads to decreased $\Delta_{47}$ values. If the reactant methane results from gas hydrate dissolution, the AOM kinetic effects may be accompanied by $\delta^{18}O$ enrichment (Fig. 6b). Future work is needed to better constrain the magnitude and direction of offset in $\Delta_{47}$ of these effects on the composition of product DIC species.

The equilibrium clumped isotope composition of $HCO_3^-$ and $CO_3^{2-}$ are thought to differ from each other, and from carbonate minerals[40,49]. Given that both pH and salinity affect

 

**Table 2 | Ancient seep carbonate geochemical data.**

| Sample | Phases | $\delta^{13}C_{carb}$ (‰,VPDB) | stdv (‰) | $\delta^{18}O_{carb}$ (‰,VPDB) | s.d. (‰) | $\Delta_{47}$ (‰, ARF) AFF 0.092 | s.e. (‰) | n |
|---|---|---|---|---|---|---|---|---|
| Tepee Buttes | | | | | | | | |
| 100-110 | Micrite | − 34.93 | 0.008 | − 10.31 | 0.0126 | 0.487 | 0.0175 | 1 |
| 130-140 | Micrite | − 37.75 | 0.007 | − 10.00 | 0.0114 | 0.432 | 0.0110 | 1 |
| 190-200 | Micrite | − 35.03 | 0.023 | − 10.64 | 0.3267 | 0.460 | 0.0185 | 1 |
| 20-30 | Micrite | − 36.40 | 0.004 | − 10.39 | 0.0056 | 0.457 | 0.0106 | 1 |
| 230-240 | Micrite | − 36.47 | 0.005 | − 7.97 | 0.0105 | 0.421 | 0.0090 | 1 |
| 250-260 | Micrite | − 35.41 | 0.002 | − 11.16 | 0.0087 | 0.389 | 0.0100 | 1 |
| 310-320 | Micrite | − 41.35 | 0.005 | − 11.07 | 0.0084 | 0.377 | 0.0138 | 1 |
| 320-2h | Micrite | − 41.05 | 0.006 | − 0.89 | 0.009 | 0.521 | 0.011 | 1 |
| 360-370 | Micrite | − 43.28 | 0.007 | − 11.09 | 0.0168 | 0.441 | 0.0075 | 1 |
| 40-50 | Micrite | − 38.01 | 0.007 | − 9.60 | 0.0089 | 0.440 | 0.0115 | 1 |
| 65-70 | Micrite | − 38.16 | 0.004 | − 11.33 | 0.0113 | 0.477 | 0.0072 | 1 |
| star 230-240 | Micrite | − 27.37 | 0.005 | − 11.10 | 0.0118 | 0.470 | 0.0138 | 1 |
| teepee hrm 210-220 | Micrite | − 40.35 | 0.006 | − 9.80 | 0.007 | 0.438 | 0.012 | 1 |
| teepee hrm 400-40 | Micrite | − 30.52 | 0.007 | − 11.17 | 0.0205 | 0.489 | 0.0122 | 1 |
| 100-110fc | Fibrous cement | − 41.22 | 0.008 | − 8.60 | 0.0111 | 0.443 | 0.0094 | 1 |
| 190-200fc | Fibrous cement | − 45.21 | 0.007 | − 7.57 | 0.0081 | 0.413 | 0.0153 | 1 |
| 210-220fc | Fibrous cement | − 45.21 | 0.005 | − 6.28 | 0.0131 | 0.388 | 0.0167 | 1 |
| 460-470fc | Fibrous cement | − 43.29 | 0.007 | − 8.09 | 0.0097 | 0.429 | 0.0085 | 1 |
| 711.5 VI S 2 | Spar | − 14.31 | | − 12.09 | | | | 1 |
| 320-2 PS | Peloidal sparite | − 44.75 | | − 0.30 | | | | 1 |
| 704.5 1 PS | Peloidal sparite | − 44.11 | | − 1.50 | | | | 1 |
| 711 WI PS | Peloidal sparite | − 37.01 | | − 4.12 | | | | 1 |
| 711.5 VI PS 2 | Peloidal sparite | − 44.76 | | − 1.04 | | | | 1 |
| 711.5 VI PS1 | Peloidal sparite | − 45.16 | 0.008 | − 1.55 | 0.013 | 0.513 | 0.012 | 1 |
| HRS 674.6 PS | Peloidal sparite | − 35.16 | | − 3.58 | | | | 1 |
| | | | | | | | | |
| Quinault Fm | | | | | | | | |
| Sample 1 Quin Fm | Micrite | − 6.05 | 0.015 | − 8.32 | 0.012 | 0.558 | 0.014 | 1 |
| Sample 9 Quin | Micrite | − 30.08 | 0.002 | 0.91 | 0.007 | 0.633 | 0.014 | 1 |
| Sample 11 Quin Fm | Micrite | − 16.43 | 0.006 | − 0.56 | 0.014 | 0.644 | 0.012 | 1 |
| | | | | | | | | |
| Pysht Fm | | | | | | | | |
| Sample 4 Pysht | Micrite | − 19.33 | 0.004 | 1.53 | 0.011 | 0.631 | 0.009 | 1 |
| Sample 5 Pysht | Micrite | − 4.03 | 0.753 | − 1.00 | 3.050 | 0.611 | 0.020 | 1 |
| | | | | | | | | |
| Panoche Hills | | | | | | | | |
| PTH 04 | Micrite | − 34.99 | 0.004 | − 0.13 | 0.009 | 0.680 | 0.011 | 1 |
| PTH-07 | Micrite | − 14.17 | 0.003 | 2.25 | 0.011 | 0.705 | 0.012 | 1 |
| PTH-08 | Micrite | − 12.53 | 0.007 | 2.25 | 0.015 | 0.682 | 0.010 | 1 |
| PTH-18 | Micrite | − 19.11 | 0.007 | − 0.71 | 0.015 | 0.671 | 0.006 | 1 |
| PTH-19 | Micrite | − 0.92 | 0.004 | − 0.18 | 0.017 | 0.649 | 0.012 | 1 |
| | | | | | | | | |
| Lincoln Creek Fm | | | | | | | | |
| Sample 8 LC | Micrite | − 15.16 | 0.006 | 1.25 | 0.012 | 0.634 | 0.008 | 1 |

DIC speciation, they can impart small but significant deviations in clumped isotope signatures of minerals (Fig. 6a)[49]. Some cold seep carbonates may have formed in environments experiencing enhanced alkalinity production within relatively saline marine waters and/or fluids influenced by clathrate formation or dissociation[50]. Theoretical modelling indicates that although the combined effects of salinity and pH affect clumped isotope values, the maximum expected differences in $\Delta_{47}$ are on the order of 0.035‰ over pH and salinity ranges of 5–14 and 0–50 parts per thousand, respectively[49]. These ranges are beyond those observed in cold seep environments. Experimental data support a somewhat larger effect with maximum offsets of ∼ 0.060‰ (ref. 40). We suggest that the effects of pH and salinity, although not sufficient enough to produce the full extent of clumped isotope anomalies observed here, may be a contributing factor given the range of fluid compositions at cold seeps.

Mixing of $HCO_3^-$ or $CO_3^{2-}$ from two different sources with distinct bulk ($\delta^{13}C$ and $\delta^{18}O$) and clumped isotope compositions does not necessarily yield a product reflecting simple linear mixing between end-members[17,51,52]. Therefore it is feasible that mixing between DIC from seawater and DIC generated from AOM could induce anomalous clumped isotope signatures in the product carbonate minerals.

In addition, geochemically heterogeneous carbonates may yield clumped isotope compositions that do not reflect true precipitation temperatures of the mixture and/or individual end-members. Defliese et al.[19] have demonstrated that mixing among carbonate phases with differing isotope compositions can produce inaccurate clumped isotope signatures and therefore have the potential to produce anomalous temperatures, such as those observed in cold seep carbonates. The degree to which the clumped isotope temperatures differ from linear mixing in a two-component system depends on the differing isotope

| Sample | Phases | Modern Temperature (°C) | $\Delta_{47}$ (‰, ARF) AFF 0.000 | $\Delta_{47}$ (‰, ARF) AFF 0.092 | s.e. (‰) | s.d. (‰) | $\Delta_{47}$ disequilibrium | |
|---|---|---|---|---|---|---|---|---|
| | | | | | | | Tang et al.[41] | Tripati et al.[40] |
| **Hydrate Ridge** | | | | | | | | |
| 2282-1c | Micrite | 5 | | 0.632 | 0.013 | | −0.121 | −0.139 |
| 2282-3 | Micrite | 5 | | 0.666 | 0.0130 | | −0.088 | −0.105 |
| 2284-1 | Micrite | 5 | | 0.661 | 0.012 | 0.028 | −0.093 | −0.111 |
| 2284-10 | Micrite | 5 | | 0.679 | 0.011 | | −0.074 | −0.092 |
| 2284-2 | Micrite | 5 | | 0.663 | 0.012 | 0.034 | −0.090 | −0.108 |
| 2284-3 | Micrite | 5 | | 0.643 | 0.0100 | | −0.110 | −0.128 |
| 2284-4 | Micrite | 5 | | 0.649 | 0.014 | | −0.105 | −0.123 |
| 2284-5 | Micrite | 5 | | 0.672 | 0.0090 | | −0.081 | −0.099 |
| 2284-6 | Micrite | 5 | | 0.659 | 0.013 | 0.039 | −0.094 | −0.112 |
| 2284-7 | Micrite | 5 | | 0.647 | 0.021 | 0.032 | −0.106 | −0.124 |
| 2284-8 | Micrite | 5 | | 0.647 | 0.0294 | | −0.107 | −0.125 |
| 2284-9 | Micrite | 5 | | 0.636 | 0.0047 | | −0.117 | −0.135 |
| CS-1-B D1 | Micrite | 5 | | 0.662 | 0.010 | | −0.091 | −0.109 |
| CS-1-B D2 | Micrite | 5 | | 0.660 | 0.005 | | −0.093 | −0.111 |
| CS-1-B L2 | Micrite | 5 | | 0.620 | 0.009 | | −0.134 | −0.152 |
| CS-1-B-L1 | Micrite | 5 | | 0.663 | 0.006 | | −0.090 | −0.108 |
| cs-1-c d1 | Micrite | 5 | | 0.629 | 0.007 | | −0.124 | −0.142 |
| CS-1-C L1 | Micrite | 5 | | 0.650 | 0.013 | | −0.103 | −0.121 |
| CS-1-C L2 | Micrite | 5 | | 0.644 | 0.010 | | −0.109 | −0.127 |
| HR-4635 3364-L-1 | Micrite | 5 | | 0.652 | 0.021 | | −0.101 | −0.119 |
| CS-1-C A1 | Fibrous cement | 5 | | 0.667 | 0.007 | | −0.087 | −0.105 |
| CS-1-C A2 | Fibrous cement | 5 | | 0.638 | 0.013 | | −0.115 | −0.133 |
| CS-1-C-A3 | Fibrous cement | 5 | | 0.663 | 0.011 | | −0.090 | −0.108 |
| HR-4635 3364-A-1 | Fibrous cement | 5 | | 0.666 | 0.012 | | −0.088 | −0.106 |
| | | | | | | | | |
| **Eel River Basin** | | | | | | | | |
| ERB-4256-G4-A | Clotted micrite | 5 | | 0.740 | 0.010 | | −0.013 | −0.031 |
| ERB-4256-G4-B | Clotted micrite | 5 | | 0.706 | 0.008 | | −0.048 | −0.066 |
| | | | | | | | | |
| **Costa Rica** | | | | | | | | |
| CR-4501-L1-CM1 | Clotted micrite | 5 | | 0.639 | 0.011 | | −0.115 | −0.133 |
| CR-4501-L1 CM 2 | Clotted micrite | 5 | | 0.657 | 0.016 | | −0.097 | −0.114 |
| CR-4502-S3 MC 1 | Micrite clast | 5 | | 0.677 | 0.009 | | −0.077 | −0.095 |
| CR 4502-L2-MC2 | Micrite clast | 5 | | 0.649 | 0.009 | | −0.104 | −0.122 |
| | | | | | | | | |
| **North Sea** | | | | | | | | |
| H03 (90C) | Clotted micrite | 1 | | 0.705 | 0.032 | | −0.063 | −0.085 |
| H03 (25C) | Clotted micrite | 1 | 0.744 | | 0.021 | 0.051 | −0.024 | −0.046 |
| H04 (90C) | Clotted micrite | 1 | | 0.733 | 0.023 | | −0.035 | −0.057 |
| H04 (25C) | Clotted micrite | 1 | 0.783 | | 0.017 | 0.018 | 0.014 | −0.008 |

**Table 3 | Degree of $\Delta_{47}$ offset from values expected from bottom water temperatures.**

compositions expressed by the two end-members. The offset is greatest when phases are most dissimilar, and mixing occurs in equal proportions (that is, a 50:50 mixture). Such mixing can produce clumped isotope compositions offset in either the positive or negative direction, yielding apparent low or high temperatures, respectively[19]. Modern and ancient cold seep carbonates commonly exhibit phase heterogeneity on 10–100 μm scales[4,11,13–15] (Fig. 3). Current clumped isotope analytical protocols preclude sampling at this resolution, therefore cold seep carbonates may yield inaccurate temperatures as a result of phase mixing.

We address the possibility of mixing-related temperature anomalies by considering bulk isotope compositions and initial formation conditions of chemical species that contribute to modern cold seep carbonates. Cold seep carbonates commonly exhibit $\delta^{13}C_{carb}$ values that extend down to ~−40 to −60‰ (Fig. 4), including those analysed here at the relatively large sample sizes required for clumped isotope quantification. We recognize the complexity of carbon cycling at the sulfate methane transition zone[53], however, as a first-order approximation, here

we assume that the $\delta^{13}C$ values represent near equal mixtures of DIC produced from AOM ($\delta^{13}C$ ranges from −60 to −100‰; in agreement with reported methane values[34,37,54–56]) and the bulk seawater DIC pool (~0‰). Some cold seep carbonates tend to exhibit $\delta^{18}O_{carb}$ values above normal marine precipitates ($\delta^{18}O_{carb}$ of ~ +5‰), consistent with precipitation from fluids associated with clathrate dissociation[32]. Maximum clumped isotope offsets are achieved when the end-members are most dissimilar[19]. In Fig. 7a we illustrate how cold seep $\delta^{13}C_{carb}$ values may relate to potential fractional contributions of methane-derived carbon ($f_{methane}$), by assuming a range of potential carbon isotope compositions of this methane ($\delta_{meth}$) and a purely marine DIC end-member (with $\delta^{13}C = 0$‰). Many modern and ancient cold seep carbonates exhibit $\delta^{13}C_{carb}$ values that could reflect equal mixtures between these two end-members ($f_{methane} = 0.5$) and therefore have the potential to express maximum mixing-related clumped isotope composition offsets.

Two mixing models are explored here to potentially reconcile the $\Delta_{47}$ anomalies (Fig. 7). The first is relevant to a marine bottom water with a temperature of ~5 °C, consistent with

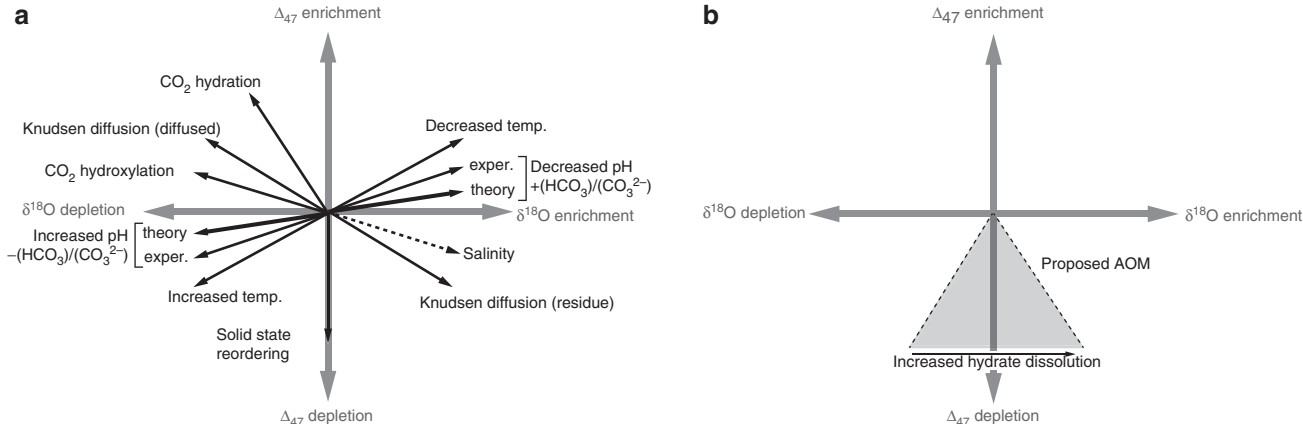

**Figure 6 | Various controls on $\Delta_{47}$ values. (a)** In addition to temperature[20], diffusion, salinity, pH and $CO_2$ hydration/hydroxylation reactions have been shown to influence the clumped isotope compositions of carbonates[40,41,45,46,49,52]. $CO_2$ hydroxylation and hydration are pH sensitive, with hydroxylation favored at higher pH[46]. Solid-state re-ordering may also impact carbonates, with increased rates at higher temperatures and greater potential for re-ordering in older rocks[22,70]. Mixing has been shown to produce variability in multiple trajectories[19]. Increased salinity produces $\delta^{18}O$ enrichment and $\Delta_{47}$ depletion, however the specific slope can vary. **(b)** Hypothesized influences by AOM reactions and hydrate dissolution.

temperatures observed at Hydrate Ridge, Eel River Basin and Costa Rica. This temperature would yield a mineral $\Delta_{47}$ value of $\sim 0.76$. The second simulates Norwegian Sea bottom waters, which exhibit temperatures of $\sim -1\,°C$ from which an equilibrium-precipitated carbonate would exhibit a $\Delta_{47}$ value of $\sim 0.77$. Mixing between more dissimilar end-members produces larger possible anomalies[19]. Here, the isotope compositions of the cold seep end-member include $\delta^{13}C$ values of $-60$, $-80$ and $-100‰$, a $\delta^{18}O$ value of $+5‰$ and $\Delta_{47}$ values of 0.71 and 0.72. The isotope compositions of the marine end-member include a $\delta^{13}C$ value of 0‰, a $\delta^{18}O$ value of 0‰ and $\Delta_{47}$ values of $\sim 0.76$ and 0.77. Mixing models between the end-members described above generally do not produce the magnitude of observed temperature disequilibrium between cold seep carbonates and modern bottom waters (Fig. 7), even though mixing yields the maximum possible offset as constrained by environmental parameters.

AOM rates are variable in modern cold seep systems (Supplementary Fig. 1). Rates commonly correspond to the flux of methane from the deeper subsurface to sulfate-containing fluids. Vent-proximal, advection-dominated methane delivery is generally associated with higher AOM rates, whereas more peripheral, diffusive seepage is generally associated with lower AOM rates[57]. As this reaction (eq. 1) produces alkalinity in equal molar proportions to the methane reactant, AOM is likely a first-order control on carbonate mineral authigenesis.

The anomalously low $\Delta_{47}$ values exhibited by modern seep carbonates may arise from rapid mineralization, which may allow them to retain an isotopic fingerprint of processes controlling solution disequilibrium, as well as surface kinetic processes. Quantitative models have shown that minerals forming rapidly will have a higher propensity to exhibit geochemical signatures out of equilibrium with the ambient environment[58], and recent observations have shown this can apply to clumped isotope signatures[40], although recent laboratory-based rapid precipitation experiments can yield equilibrium clumped isotope signatures[59]. In cold seep settings, mineral precipitation rates are difficult to quantify, so that although there are many reports of AOM rates, authigenic carbonate precipitation rates are rarely available. Where carbonate precipitation and AOM rates have been quantified in the Nile Delta[60,61] and Mediterranean Sea[62,63] precipitation rates broadly overlap with AOM rates (see Supplementary Fig. 1). Therefore environments exhibiting

more rapid AOM rates may also exhibit more rapid carbonate precipitation rates and carbonates expressing more severe disequilibrium behaviour.

The above combination of factors likely explains the large disequilibrium signatures observed in modern seep carbonates. These same processes would be relevant in ancient seeps, and therefore it is reasonable to assume that ancient seeps also exhibit disequilibrium. The ancient seep carbonates measured here all show evidence for apparent temperatures that are too warm to reflect bottom waters. On the one hand, this is consistent with our explanation for modern seep disequilibrium, however, on the other hand, ancient deposits are generally composed of many distinct phases, some perhaps precipitated from warm, later diagenetic fluids[11,13].

We suggest that although modern samples record disequilibrium (that is, non-temperature dependent) $\Delta_{47}$ values, ancient samples may have experienced subsequent heating, solid-state re-ordering (Fig. 6) or both that may overprint the signal and randomize $^{13}C$–$^{18}O$ bonds from an initial ordering. Complex parageneses have been interpreted for ancient cold seep carbonate cement relationships[11,14,15,38]. Thus in an ancient seep setting, extremely low $\Delta_{47}$ values could be recorded. In fact, it has been argued that Neoproterozoic Doushantuo carbonates exhibit low $\Delta_{47}$ values consistent with hydrothermal conditions[51]. However, based on our modern and ancient seep data, an alternative possibility is that the Doushantuo carbonates exhibited substantial disequilibrium $\Delta_{47}$ signatures on initial formation, and that subsequent processes (for example, solid-state re-ordering[22]) have randomized the signatures from original values. Evidence to support this interpretation stems from our fossil seep carbonates of the Tepee Buttes, which show clumped isotope and $\delta^{18}O_{carb}$ values more similar to those of Doushantuo carbonates[51] than modern seep carbonates.

In conclusion, modern cold seep carbonates exhibit $\Delta_{47}$ values ranging from 0.609 to 0.783‰, significantly lower than equilibrium precipitation would yield. This disequilibrium translates into apparent formation temperatures of $-3$ up to $52\,°C$, while modern, seep-proximal bottom waters are all $<5\,°C$. We hypothesize kinetic effects likely arise from a combination of processes, including AOM, DIC speciation effects and mixing, with the rapid precipitation rates in such settings leading to the high potential for incorporation of such kinetic isotope signals into carbonate minerals. In ancient cold seeps these same factors,

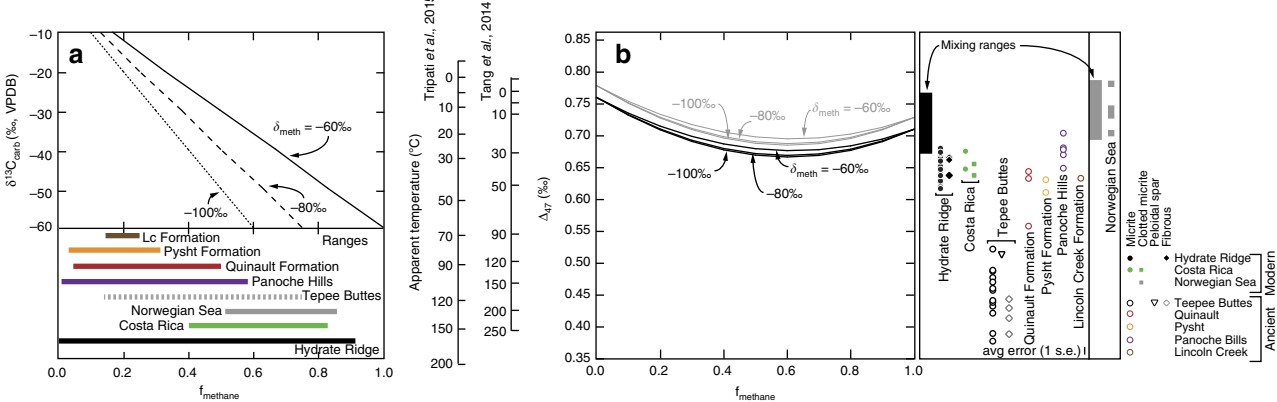

**Figure 7 | Comparison of clumped isotope compositions with mixing-related temperature anomalies.** (**a**) Diagram depicting the fractional contribution of methane-derived carbon ($f_{methane}$) based on $\delta^{13}C_{carb}$ values and a range of methane carbon isotope compositions ($\delta_{meth}$). Here, $f_{methane}$ is calculated assuming that carbon is derived from methane and seawater (with $\delta^{13}C = 0$). Ranges of $f_{methane}$ arise from variable $\delta^{13}C_{carb}$ and hypothetical $\delta_{meth}$ values. Notice how modern and most ancient seep carbonate $\delta^{13}C_{carb}$ values overlap with those required for a 50:50 mixture (that is, $f_{methane} = 0.5$). (**b**) Results of mixing between two environmentally constrained end-members as calculated using the approach of Defliese et al.[65] Grey envelope and curves correspond to the potential range of values resulting from mixing in Norwegian Sea carbonates. Black envelope and curves correspond to all others. See text for information concerning end-member isotope compositions. Notice that mixing cannot account for the clumped isotope and temperature anomalies recorded in cold seep carbonates, both the Tripati et al.[40] and Tang et al.[41] calibration-derived temperatures included. Note that dolomites of the Eel River Basin are not included in this diagram. This is because positive $\delta^{13}C_{carb}$ values indicate formation in the zone of methanogenesis and therefore potential mixing cannot be independently assessed as with the other samples.

in combination with burial diagenesis and/or solid-state reordering, can result in extremely low $\Delta_{47}$ values. Ultimately, our findings indicate that cold seep carbonates formed by AOM are susceptible to disequilibrium behaviour and therefore clumped isotopes cannot be used as a strict temperature proxy for the primary formation environment in modern or ancient seeps.

## Methods

**Sample collection.** Carbonate samples were collected from modern cold seeps of Hydrate Ridge, off the western coast of Costa Rica, Eel River Basin and the Norwegian Sea (Fig. 1). Micritic and fibrous carbonate phases were mirodrilled from these modern cold seep carbonates. Ancient cold seep carbonates were collected from outcrops of the Quinault, Pysht and Lincoln Creek Formations, the Tepee Buttes and the Panoche Hills (Fig. 1). In addition to micritic and fibrous phases, ancient carbonates also exhibit sparry phases[11,14,15,38].

**Sample preparation and analyses.** Between 8 and 14 mg of powdered sample was used for each measurement. The powders were pretreated in 30%, cold ($\sim 10\,^\circ C$) hydrogen peroxide to mitigate potential contamination by organic compounds. Some samples were measured in duplicate to evaluate reproducibility. Sample powders were digested in phosphoric acid at 25 or 90 $^\circ C$ (Table 1) and analysed following published protocols[64]. The resulting $CO_2$ was analysed on a customized Thermo MAT 253 gas source mass spectrometer at UCLA dedicated to measuring clumped isotopes in $CO_2$. A custom-built, automated, online device[64] is used to introduce samples to the mass spectrometer.

**Isotope data.** Values of $\Delta_{47}$ are reported as the per mil (‰) difference relative to what would be predicted given a stochastic distribution of isotopes among all possible isotopologues. Values of $\Delta_{47}$ are determined by calculating $\delta^{18}O$, $\delta^{13}C$, and the abundance of mass-47 isotopologues[52]. This spectrum of masses along with known fractionation factors for $^{18}O/^{16}O$ and $^{13}C/^{12}C$ allows for quantification of carbonate $\delta^{18}O$, $\delta^{13}C$ and $\Delta_{47}$ values[20]. Data are shown in Tables 1 and 2. All $\Delta_{47}$ values are reported on the absolute reference frame (ARF), which is calculated using 25 and 1,000 $^\circ C$ equilibrated gases. These measurements were made before the implementation of a baseline correction scheme.

These $\Delta_{47}$ values were converted to temperature using various calibrations[20,40,41,65–68]. Figures and discussion are based on the calibrations by Tang et al.[41] and Tripati et al.[40] and acid fractionation factor of 0.092‰ (ref. 69), which is within error of the value reported by Defliese et al.[65] The acid digestion temperatures, calibrations and fractionation factors yield comparable temperatures (aside from the Dennis and Schrag[66] calibration, which significantly lower and produces some impossible temperatures). Carbonate $\delta^{13}C$ and $\delta^{18}O$ values are reported in ‰ relative to the VPDB standard. The average $\Delta_{47}$ precision based on measurements of carbonate standards and samples during the runs for this study

was 0.012‰ (1 s.e.), and select replicate analyses yield an average standard deviation of 0.033‰ (1 s.d.). The average reproducibility of carbonate $\delta^{13}C$ and $\delta^{18}O$ are 0.006 and 0.011‰, respectively (Tables 1 and 2).

**Data availability.** All data pertinent to this manuscript and its reported findings can be found in the manuscript itself or the associated Supplementary Information file.

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

## Acknowledgements

Preliminary discussion in access to some cold seep carbonates provided by Victoria Orphan. Funding for the study was provided to S.L. by the Agouron Geobiology Post-doctoral Fellowship. A.K.T. acknowledges support from the Department of Energy through BES grant DE-FG02-13ER16402.

## Author contributions

S.J.L and A.E.T conceived study; S.J.L, R.E.T, K.B., L.G.H. and A.E.T performed analyses; J.S., M.H., M.T., J.M. and T.L. provided samples. All authors contributed to writing the manuscript.

## Additional information

**Competing financial interests:** The authors declare no competing financial interests.

