## [Peer Review File · Nature Communications]

Reviewers' Comments:

Reviewer #1 (Remarks to the Author):

This is a nice paper which presents evidence of non-equilibrium D47 precipitation. I have no major criticisms except for a few minor suggestions on the paper.

Reviewer #2 (Remarks to the Author):

Data are presented on the carbon, oxygen, and clumped isotope composition of carbonates from methane cold seeps in modern and ancient environments. The main result is that the clumped isotope compositions do not recover the measured temperatures of carbonate formation in the modern seeps, and therefore cannot be used to reconstruct temperatures of formation in ancient deposits. This is essentially a negative result.

The manuscript initially leads the reader to believe that temperature is the primary objective:

"A particularly useful parameter to quantify is precipitation temperature, as it dictates thermodynamic considerations such as abiotic versus biotic reaction times, and gas hydrate dynamics. The newly emerging clumped isotope proxy has shown promise as a powerful geothermometer in the geosciences, yet the utility of clumped isotopes as an accurate geothermometer in cold seep carbonates has yet to be demonstrated."

But oddly, this is followed by:

"Here, we explore the possibility of non-temperature dependent carbonate clumped isotope signatures in cold seep carbonates through analyses of modern precipitates forming under well-constrained conditions (i.e., temperatures, pH, salinities and fluid $\delta^{18}\text{O}$ values)."

I doubt the possibility of non-temperature dependent clumped signatures was the original motivation behind this study. Nevertheless, such effects have received a lot of attention lately in other systems such as speleothem carbonates, coral carbonates and O_2 in the atmosphere. The text goes on to discuss various processes that could lead to clumped isotope disequilibrium, such as rapid precipitation, mixing of DIC pools, and/or precipitation from a DIC pool that is not equilibrated due to hydration-hydroxylation reactions or AOM oxidation. All of these processes have been discussed in the literature (too many references to list here) and so there really isn't anything novel about the proposed mechanisms. Overall, the conclusions are rather uninspiring because (i) the data do not shed new light on underlying mechanisms of kinetic clumped isotope effects that haven't been discussed elsewhere, (ii) the data do not provide new constraints on the sign and magnitude of disequilibrium $\Delta 47$ associated with any of the aforementioned processes that haven't been discussed elsewhere, (iii) there is nothing in the discussion that points to improved predictive capabilities for clumped isotope disequilibrium, and (iv) the insights presented cannot be extended other systems where clumped isotope disequilibrium is also observed. So, although the data are

certainly valuable insofar as they add to the compendium of natural observations of clumped isotope disequilibrium, I don't see how the paper will significantly influence the current thinking of equilibrium vs. disequilibrium in the field of clumped isotope geochemistry.

Reviewer #3 (Remarks to the Author):

This article presents clumped isotope analyses of carbonate concretions, formed due to supersaturation caused by alkalinity production from the anoxic oxidation of methane (AOM). They looked at multiple occurrences of this type of deposit, and all modern examples indicate formation temperatures of ~ 50 °C or more above the in situ temperature. The significance of this work is that clumped isotope analysis is a new technique for estimating carbonate formation temperature. Its applicability is just being explored, and this article demonstrates some important limitations of the approach. The article represents an extension of laboratory experiments to evaluate the role of environmental variables (pH, salinity, growth rate), published by Tripathi et al (2015). The present study examines field evidence about whether this type of carbonate might be formed in equilibrium with the surrounding fluid. Rapid precipitation kinetics are the likely cause of the disequilibrium observed in these AOM settings, although this cannot be established definitively. The article is focused, clearly written, and well illustrated.

Data appear to be of high quality, statistics are adequate, and the conclusions are warranted based on this data set. Citations appear to be adequate.

I strongly recommend publication, as this will serve as a benchmark for further studies of this kind, an approach that is growing.

A few minor points:

Line 210: Actually, they cannot fully discount this, as they would need to know the fluid composition to really discount this mechanism.

Line 255: Replace "and" with "or"

Line 340: Insert "formed by AOM" before "are"

Line 351: More detail about the peroxide treatment would be helpful. How strong is the solution, was leaching done after powdering, what temp was used, etc.

Line 416: correct "ancient"

Fig. 5: Place the avg error label and symbol inside the top of the box above, so that readers do not try to read this as an axis label.

Fig. 6: This figure is ok, but has only moderate utility in my view.

Fig. 7 and discussion on page 13, paragraph 2. I could not understand how the end-member mixing equation was formulated to produce the curvature in panel B. This should be clarified, as the formation of curvature from mixing is not intuitive.

Author Response to Review

Please find responses to the second round of review below. Included line numbers indicate changes to the text in the marked draft, where appropriate. Reviewer comments are provided in regular text, responses are provided in italics. Note that the marked manuscript draft also contains the style edits provided by the Associate Editor.

Reviewer #1 (Remarks to the Author):

This is a nice paper which presents evidence of non-equilibrium D47 precipitation. I have no major criticisms except for a few minor suggestions on the paper.

Changes have been made as outlined in the attached document. Lines 100, 110, 118, 119, 140, 149-150, 156, 199-200, 219 and Figure 5 have been modified.

Reviewer #2 (Remarks to the Author):

Data are presented on the carbon, oxygen, and clumped isotope composition of carbonates from methane cold seeps in modern and ancient environments. The main result is that the clumped isotope compositions do not recover the measured temperatures of carbonate formation in the modern seeps, and therefore cannot be used to reconstruct temperatures of formation in ancient deposits. This is essentially a negative result.

The manuscript initially leads the reader to believe that temperature is the primary objective:

"A particularly useful parameter to quantify is precipitation temperature, as it dictates thermodynamic considerations such as abiotic versus biotic reaction times, and gas hydrate dynamics. The newly emerging clumped isotope proxy has shown promise as a powerful geothermometer in the geosciences, yet the utility of clumped isotopes as an accurate geothermometer in cold seep carbonates has yet to be demonstrated."

But oddly, this is followed by:

"Here, we explore the possibility of non-temperature dependent carbonate clumped isotope signatures in cold seep carbonates through analyses of modern precipitates forming under well-constrained conditions (i.e., temperatures, pH, salinities and fluid $\delta^{18}\text{O}$ values)."

I doubt the possibility of non-temperature dependent clumped signatures was the original motivation behind this study. Nevertheless, such effects have received a lot of attention lately in other systems such as speleothem carbonates, coral carbonates and O_2 in the atmosphere. The text goes on to discuss various processes that could lead to clumped isotope disequilibrium, such as rapid precipitation, mixing of DIC pools, and/or precipitation from a DIC pool that is not equilibrated due

to hydration-hydroxylation reactions or AOM oxidation. All of these processes have been discussed in the literature (too many references to list here) and so there really isn't anything novel about the proposed mechanisms. Overall, the conclusions are rather uninspiring because (i) the data do not shed new light on underlying mechanisms of kinetic clumped isotope effects that haven't been discussed elsewhere, (ii) the data do not provide new constraints on the sign and magnitude of disequilibrium $\Delta 47$ associated with any of the aforementioned processes that haven't been discussed elsewhere, (iii) there is nothing in the discussion that points to improved predictive capabilities for clumped isotope disequilibrium, and (iv) the insights presented cannot be extended other systems where clumped isotope disequilibrium is also observed. So, although the data are certainly valuable insofar as they add to the compendium of natural observations of clumped isotope disequilibrium, I don't see how the paper will significantly influence the current thinking of equilibrium vs. disequilibrium in the field of clumped isotope geochemistry.

Whereas the reviewer feels our results do not shed light on the mechanism of disequilibrium signal inheritance, the major finding that cold seep carbonates do not exhibit equilibrium signatures in modern settings is useful to the broader geoscience community and deserves publication. These carbonates (and cold seep systems in general) are the subject of intense debate owing to their association with biogeochemical cycling, mineralization and greenhouse gas consumption. Our ability to characterize ancient seep systems relies heavily on proxy-based records and we herein demonstrate that clumped isotope signals do not reflect equilibrium compositions and therefore cannot be used as a straightforward paleothermometer.

Reviewer #3 (Remarks to the Author):

This article presents clumped isotope analyses of carbonate concretions, formed due to supersaturation caused by alkalinity production from the anoxic oxidation of methane (AOM). They looked at multiple occurrences of this type of deposit, and all modern examples indicate formation temperatures of ~ 50 °C or more above the in situ temperature. The significance of this work is that clumped isotope analysis is a new technique for estimating carbonate formation temperature. Its applicability is just being explored, and this article demonstrates some important limitations of the approach. The article represents an extension of laboratory experiments to evaluate the role of environmental variables (pH, salinity, growth rate), published by Tripathi et al (2015). The present study examines field evidence about whether this type of carbonate might be formed in equilibrium with the surrounding fluid. Rapid precipitation kinetics are the likely cause of the disequilibrium observed in these AOM settings, although this cannot be established definitively. The article is focused, clearly written, and well illustrated.

Data appear to be of high quality, statistics are adequate, and the conclusions are warranted based on this data set. Citations appear to be adequate.

I strongly recommend publication, as this will serve as a benchmark for further studies of this kind, an approach that is growing.

A few minor points:

Line 210: Actually, they cannot fully discount this, as they would need to know the fluid composition to really discount this mechanism.

Lines 246-247 have been weakened.

Line 255: Replace "and" with "or"

Line 296 modified.

Line 340: Insert "formed by AOM" before "are"

Line 383 modified.

Line 351: More detail about the peroxide treatment would be helpful. How strong is the solution, was leaching done after powdering, what temp was used, etc.

Line 405 modified.

Line 416: correct "ancient"

Line 498 modified.

Fig. 5: Place the avg error label and symbol inside the top of the box above, so that readers do not try to read this as an axis label.

Figure modified.

Fig. 6: This figure is ok, but has only moderate utility in my view.

We feel this figure provides a straightforward visualization of the proposed disequilibrium signatures incorporated in cold seep carbonates.

Fig. 7 and discussion on page 13, paragraph 2. I could not understand how the end-member mixing equation was formulated to produce the curvature in panel B. This should be clarified, as the formation of curvature from mixing is not intuitive.

The references provided in this paragraph discuss the calculations in detail. With the short format style of Nature Communications, there is not enough room for reiteration.

In addition to the edits suggested by the referees, the Associate Editor has suggested stylistic edits. These specific edits are indicated below.

Line 26 modified.

Line 82 (last paragraph in introduction) modified.

Line 88 (Results Section Begins) modified.

Lines 95, 144 (removal of subheadings) modified.

Lines 220, 222 (Table 4 Changed to Supplementary Table 1) modified.

Line 375 (conclusion header removed) modified.

Lines 388, 396, 413 (addition of methods subheadings) modified.

Lines 435-437 (data access statement) added.

Line 456 (Figure 1 title separated from caption) modified.

Lines 460-464 (Figure 2 caption edited, expanded) modified.

Lines 466-474 (Figure 3 caption edited, expanded) modified.

Line 477 (Figure 4 title separated from caption) modified.

Line 505 (Figure 5 title separated from caption) modified.

Line 511 (Figure 6 title separated from caption) modified.

Lines 512, 518 (lower case a and b inserted) modified.

Line 529-530 (Figure 7 title separated from caption) modified.

Lines 531, 536 (lower case a and b inserted) modified.

Figures removed from text file.

Table 4 removed from main text, provided as part of the Supplemental File.